# Simulation design to find the welfare impacts of livestock trading and disease transmission

**Hyeonjun Hwang** [ORCID]*

Graduate School of Data Science, Kyungpook National University, Daegu, South Korea

* hhwang@knu.ac.kr

## Abstract

This study designs a theoretical model and simulation model that can explain the welfare impacts of disease transmission that occurs in livestock trade. A household production model and a SIR model are used to find theoretical profitable conditions for infectious livestock trading and prices and quantities for transactions. Under the theoretical conditions an agent-based model is used to simulate livestock transactions to compare social impacts based on the number of livestock, household wealth and income, and wealth inequality. Asymmetric information is used to assign tendencies of livestock trading agents. Buyers are assumed to be uninformed about the health status of livestock owned or used by sellers, while sellers are either uninformed for their herd's health status, and if informed, the sellers' behavior of selecting infectious livestock for transactions is divided into selfish selection and altruistic selection. The simulation results reveal that livestock losses are higher when trading occurs, but overall economic welfare tends to increase with trade. Interestingly, when sellers selfishly sell sick animals, average household wealth and income peak, albeit with greater wealth inequality.

**Data Availability Statement:** All relevant data are within the manuscript.

**Funding:** Kyungpook National University Research Fund, 2022. The funders had no role in study design, data collection and analysis, decision to publish, or preparation of the manuscript.

## Introduction

Smallholder farmers heavily rely on livestock markets as a pivotal platform for buying and selling animals, integral to their herd management and husbandry practices. However, it is important to acknowledge that these market transactions can inadvertently serve as a conduit for the transmission of infectious diseases across different herds. The consequences of such disease transmission are far-reaching, contributing to the endemic prevalence of diseases within communities and even having the potential to spark epidemics [1–3]. Research has established that livestock markets play a significant role in facilitating the spread of a wide spectrum of diseases. Examples include the transmission of diseases such as foot and mouth disease among cattle [4] and classical swine fever [5]. The movement of livestock between different farms amplifies the probability of infected animals being transferred [6]. In the context of pig farming, for instance, the movement of pigs between farms markedly heightens the risk of introducing infectious diseases like foot and mouth disease, a phenomenon that is evident in both Europe [2, 7, 8] and the United States [9]. Furthermore, within the cattle trade process, the dynamics of infectious disease spread are discernible [4, 10]. These dynamics underscore the

**Competing interests:** The author has declared that no competing interests exist.

intricate relationship between livestock markets, disease transmission, and their potential repercussions on both animal health and public health.

In communities heavily reliant on livestock, the economic consequences of infectious disease transmission can be devastating. Consider the case of foot and mouth disease, where the direct economic losses manifest in several ways. These losses encompass decreased production rates [11] and a decline in the growth rate of livestock herds [12]. The consequences of such an outbreak are particularly pronounced in impoverished regions that are heavily dependent on livestock for their livelihoods. The direct repercussions of infectious diseases, such as foot and mouth disease, can be severe. They include a notable reduction in milk production and diminished herd fertility. These impacts are especially acute in economically disadvantaged areas where the local population relies heavily on livestock for their sustenance and income [13]. In essence, the transmission of livestock diseases within impoverished rural areas, where livestock production plays a vital role, can lead to detrimental economic effects. These effects not only impair the well-being of individual households but also have broader implications for the overall economic stability and development of such communities.

In the context of infectious disease transmission through livestock market transactions, a significant issue arises from the presence of information asymmetry between sellers and buyers. Information asymmetry refers to an imbalance in the knowledge and information available to buyers and sellers, which can lead to inefficient outcomes within markets. Numerous studies have explored the challenges posed by this asymmetric information problem [14–17], and some noteworthy examples are worth mentioning. One notable application of the concept of asymmetric information in the livestock market is the dominance of Holstein breed sales in cattle auctions [18]. This phenomenon can be attributed to the information gap between buyers and sellers, where buyers may lack adequate information about the cattle they are purchasing, including details about their breed, health status, or vaccination history. In the context of cattle auctions, the presence of information asymmetry can also result in additional costs, such as re-vaccination expenses, which occur when buyers lack essential information about whether the cattle being sold by sellers have been vaccinated against specific diseases [19]. Overall, the issue of information asymmetry in livestock markets underscores the need for mechanisms or interventions to address this imbalance, ultimately promoting more efficient and equitable transactions within these markets.

There are two scenarios in which a seller can inadvertently sell infected livestock, depending on the seller's perception of the health information within their own farm. In both cases, it is assumed that the buyer is always uninformed about the health status of the livestock being sold by the sellers. In the first scenario, if a seller is uninformed about which of their livestock has an infectious disease, the disease can be transmitted to a buyer when the purchased livestock, randomly selected from the seller's herd, happens to have the disease [20]. This first scenario is referred to as the "Uninformed Seller" case in this study. In the second scenario, even when a seller possesses precise knowledge about the health status of their livestock, the seller intentionally sells infectious livestock because they believe that livestock trading is a profitable endeavor [21]. This situation is labeled as the "Informed Seller" case. If the livestock transferred to the buyer's farm through livestock trading subsequently triggers an epidemic within the buyer's herd, the seller can be categorized as either the uninformed case or the informed case, depending on their level of knowledge and intention in selling infectious livestock.

In the realm of livestock transactions characterized by asymmetric information regarding livestock health, there has been comparatively less attention given to the economic implications of infectious disease transmission. The problem of asymmetric information related to infectious livestock diseases has primarily been examined in the context of interactions between farmers who identify sick animals and government agencies responsible for disease

control. From an economic perspective, when a farmer detects a livestock disease outbreak within their herd, they face a decision-making process. This involves comparing the benefits derived from government-funded subsidies for quarantine and the disposal of livestock with the costs associated with self-reporting the disease. The farmer must choose whether to fully comply with government control measures or opt for personal disposal of the affected livestock [21–24]. Utilizing a principal-agent model involving both the government and farmers, it becomes evident that farmers require adequate compensation to incentivize them to self-report infectious livestock cases, thus preventing the spread of the disease [20]. It is noteworthy that in the references mentioned, the focus has primarily been on the decision-making process of farmers when they detect livestock diseases within their herds. However, these discussions have not delved into the economic consequences that arise when sick animals are actually used in trading. Moreover, they have not explored the re-vaccination costs borne by buyers nor presented results related to the overall social well-being influenced by market transactions involving sick animals [19]. These aspects highlight the need for a more comprehensive examination of the economic ramifications of infectious disease transmission within livestock markets.

This study introduces an Agent-Based Model (ABM) to simulate the dynamics of inter-household livestock trading within an environment affected by infectious diseases. The primary objective is to investigate the welfare implications of trading activities among producer-households when confronted with incomplete and asymmetric information. Households under consideration commence with an endowment of wealth, comprised of livestock assets and a generic riskless liquid asset. These assets generate income, which constitutes the foundation for household consumption decisions. A significant motivation for inter-household livestock trading in this model is the variable disease mortality rates observed among individual household herds. This inherent variability provides a compelling incentive for households to engage in trading, as it enables them to diversify risks associated with disease-related losses. However, the act of trading itself introduces an additional layer of risk, as it carries the potential for further disease transmission among herds. To assess the welfare effects of livestock trading within this infectious disease context, a series of key metrics are employed. These metrics include the quantification of "Total Livestock," which serves as a measure of the aggregate livestock population within the model. This metric aids in evaluating the overall health and size of the livestock population under varying trading and disease transmission scenarios. Furthermore, "Household Consumption Levels" are rigorously examined to gauge the impact of trading activities and ensuing disease transmission on the well-being and standard of living of individual households. Lastly, "Gini Coefficients for Wealth" are computed for both liquid assets and livestock holdings. These Gini coefficients provide a comprehensive analysis of wealth distribution among households, thereby shedding light on potential disparities that may emerge as a consequence of trading and disease transmission.

In addition to these aspects, this study explores scenarios with homogeneous or heterogeneous initial herd sizes and considers scenarios in which buyers and sellers are either symmetrically uninformed about the disease status of traded animals or the seller is better informed and either chooses to selectively sell healthy or sick animals. Given our model parameterizations, I find, in general, that gains from trade outweigh trade-induced livestock value losses due to trade-induced disease transmission. Additionally, I observe that when sellers selfishly sell sick animals to less-informed buyers, the total aggregate livestock and wealth are lower than when sellers are altruistic or symmetrically uninformed about animal health status. Among other findings, I also note that the more similar the initial herd size is between households, the greater the cumulative social well-being through increased consumption. This comprehensive analysis offers valuable insights into the intricate relationship between livestock trading, disease transmission, household welfare, and herd characteristics.

## Theoretical model

Agent-based modeling (ABM) represents a computational approach wherein mathematical behavioral rules define computational agents, guiding their decisions and interactions. In this section, I establish a foundational theoretical model from which these behavioral rules are derived. The process initiates with the formulation of a household production and consumption model, which yields the essential general equilibrium conditions governing livestock trading. Subsequently, I introduce disease dynamics and elucidate their interplay with household production and consumption patterns. This conceptual framework allows us to delineate the welfare metrics employed to evaluate the economic implications of livestock trading within an infectious disease context, a phenomenon that I simulate within the ABM model. Through this approach, I endeavor to gain comprehensive insights into the multifaceted dynamics of inter-household livestock trading while considering the influence of infectious diseases.

### Household production

Livestock trading among households is governed by the determination of a general equilibrium price, which stems from utility maximization utilizing the household production framework. The household production model [25] encompasses both production and consumption activities within households. In the household production model households are both consumers and suppliers themselves, so both profit maximization and utility maximization are considered. It is noteworthy that, unlike other household production models, this study does not factor in wages for self-employment; instead, it exclusively considers profits derived from livestock production as the primary income source. Outputs related to livestock, such as milk and meat, are produced by households for internal consumption. Through the individual household production model and adherence to market-clearing conditions, I derive the trading volume and equilibrium price governing inter-household transactions. This approach allows us to quantitatively ascertain the dynamics of livestock trading within a broader economic context, emphasizing the role of household production and consumption patterns in shaping market outcomes.

This research focuses its investigation on households heavily reliant on livestock as their sole asset and primary means of production. Each household, denoted as $i$ possesses a herd of generic livestock, represented as $L_{it}$ in each discrete time period $t$ where $L_{it}$ generates a generic physical product, $F(L_{it})$ governed by the single input production function [26]

$$F(L_{it}) = \alpha \ln(L_{it}) \quad \text{where} \quad \alpha \in (0, 1) \tag{1}$$

where $i$ indexes the individual households ($i = 1, 2, \cdots, N$), while $t$ indexes the discrete time periods $t = 0, 1, 2, \cdots, T$. An important assumption underpinning this study is that livestock health status does not influence household production output. In other words, the illness of livestock solely results in a change in the number of animals within a herd, without affecting the productivity per live animal. Livestock health is categorized into two groups: healthy livestock (comprising susceptible (S) and recovered (R) livestock, as per the SIR model) and unhealthy infectious livestock (I). Since the total livestock population $L = S + I + R$ is given by the sum of susceptible (S), infectious (I), and recovered (R) livestock in Eq (1), there is no distinction in productivity among these categories (S, I, and R).

In addition to their livestock holdings, households also possess a liquid asset denoted as $A_{it}$. This liquid asset can be utilized for purchasing livestock or the consumption good $C$ and yields a return of $r$ per discrete time period. Conceptually, this asset operates akin to a savings account, accruing interest at a rate of $r$. Households commence at time $t = 0$ with initial endowments of livestock, denoted as $L_{i0}$ (where $L_{i0} > 0$), and liquid assets, denoted as $A_{i0}$

(where $A_{i0} > 0$). It is important to note that this study assumes separability between household consumption and production activities, implying that households make decisions regarding production and asset allocation to maximize net income (profit). Subsequently, this income is employed to maximize utility [25, 27, 28]. The household profit function, denoted as $\pi_{it}$, is expressed as follows:

$$\pi_{it} = F(L_{it} + X_{it}) + rA_{it} - p_t X_{it} \qquad (2)$$

where $p_t$ represents the inter-household transaction price for livestock trade at time $t$, and $X_{it}$ signifies the net purchase of livestock. The price of the livestock product $F(L_{it})$ is the numeraire, with a price set to unity. This formulation captures the dynamics of household profit generation, reflecting both production and asset allocation decisions, within the context of livestock trading and financial management.

In the face of livestock disease mortality, households are confronted with the necessity of making strategic asset allocation decisions, involving the purchase or sale of livestock at prevailing market prices. In some instances, households may choose to utilize a portion of their available liquid asset, denoted as $A_{it}$, for livestock acquisitions. To streamline the computational complexity within the Agent-Based Model (ABM), I adopt a simplifying assumption. Specifically, I assume that household consumption ($C_{it}$) is contingent solely upon the current income generated by their existing assets, represented as $F(L_{it} + X_{it}) + rA_{it}$. Consequently, household consumption remains stable and unaltered by short-term livestock transactions, which might either deplete or augment the liquid asset $A_{it}$:

$$C_{it} = F(L_{it} + X_{it}) + rA_{it}. \qquad (3)$$

In practical terms, this means that households do not alter their consumption patterns in response to windfalls resulting from livestock sales, nor do they curtail consumption when acquiring livestock by drawing from their savings. Instead, they adopt a consumption-smoothing strategy, wherein they consume only what is sustainable based on their current income derived from the combined assets. This approach aligns with established economic concepts, such as the permanent income hypothesis, and adheres to standard dynamic models that advocate for consumption smoothing due to diminishing marginal utility from consumption.

The decision-making process of a household involves separate considerations for the profits generated from livestock trade and the interest accrued from liquid assets. At each time period $t$, the household engages in a consumption decision, determining how to allocate the net returns. The household's objective is to maximize personal utility through the optimal allocation of resources. To represent the single good used for livestock product consumption, denoted as $C_{it}$, I assume the adoption of a logarithmic utility function. This assumption reflects the concept of diminishing returns to consumption, whereby increasing consumption yields diminishing marginal utility. This modeling choice aligns with established economic principles and allows us to capture the household's utility-maximizing behavior within the context of livestock trade and consumption decisions.

$$U(C_{it}) = \ln C_{it} \qquad (4)$$

Then, the the consumption decision is from the utility maximization such as

$$\max_{C_{it}} \ln C_{it}$$

$$\text{s.t.} \quad C_{it} = \alpha \ln(L_{it} + X_{it}) + rA_{it}$$

$$A_{it} = A_{i0} - \sum_{\tau=1}^{t} p_\tau X_{i\tau} \tag{5}$$

$$p_t X_{it} \leq A_{it} \quad \text{if} \quad X_{it} > 0$$

$$\forall i, A_{i0} = A_0 \quad \text{given.}$$

In the optimization problem outlined in Eq (5), several constraints guide the allocation of resources. The first constraint ensures that consumption at time $t$ ($C_{it}$) is bounded by the net returns derived from two income sources at that time: production returns ($A \ln(L_{it} + X_{it})$) and interest returns ($rA_{it}$). The second constraint specifies that changes in liquid assets over time are solely influenced by livestock trading activities. The third constraint limits the payment for livestock transactions by a household ($i$) to its available liquid assets at time $t$. In other words, a positive net purchase means that a household is purchasing livestock, and it is impossible to purchase more than the assets of the entire household. If net purchase is negative, this means that livestock is being sold, and there are no restrictions based on assets. Lastly, the fourth constraint establishes that all households receive an identical initial endowment, maintaining equality at the outset of the model. These constraints govern the dynamic resource allocation decisions of households, encompassing production returns, interest income, livestock trading, and the equitable distribution of initial assets.

The utility maximization problem represented by Eq (5) presupposes that households make immediate, constrained decisions driven by the prevailing needs at time $t$. These decisions are myopic in nature, meaning they prioritize the current moment and do not consider the broader, long-term livestock dynamics. This myopic approach is adopted due to the computational complexity associated with a full dynamic programming problem, which can be resource-intensive. As a result of this assumption, information pertaining to past or future livestock transactions has no bearing on the current inter-household transactions. In essence, individual households' consumption decisions are made in isolation from temporal considerations, and the condition for clearing the livestock market is solely determined by the volume of livestock trading transpiring during the current time period $t$.

In the livestock-dependent society under consideration, general equilibrium prices are determined to equate the demands and supplies of individual households. Transactions within this closed economy are executed for a single time period in accordance with these equilibrium prices. It is assumed that livestock transactions occur exclusively within this controlled environment, with no involvement of external markets. However, non-market transactions among households within the livestock-dependent community are permitted, thereby allowing for the exchange of outcomes produced by each household. Given the absence of external markets, the transaction mechanism within this pure-exchange economy leads to the establishment of market clearing conditions in each time period $t$ within the livestock-dependent society. These conditions ensure that the supply and demand for livestock are balanced within the confines of this self-contained economic system:

$$\sum_{i=1}^{N} X_{it} = 0$$

which means the establishment of general equilibrium prices implies that the total demands of

consumers (denoted as $X_{it} < 0$) and the sales made by sellers ($X_{it} > 0$) are entirely balanced through transactions taking place within this society at time $t$. Put differently, the desire for livestock trading within the society during time $t$ must be exactly matched by the supply offered by other households. This equilibrium condition ensures that there is no excess demand or supply in the livestock market, fostering a state of equilibrium where the transactions of livestock are efficiently matched within the society.

The livestock (excess) demand, stemming from the utility maximization process utilizing household production, and the corresponding livestock price determined by the market clearing conditions are as follows.

$$X_{it}(p_t, L_{it}) = \frac{\alpha}{rp_t} - L_{it} \tag{6}$$

$$p_t(N, L_{it}) = \frac{\alpha N}{r \sum_{i=1}^{N} L_{it}}. \tag{7}$$

The equilibrium price of livestock trading at time $t$, denoted as $p_t$, is determined by several key factors, including the total number of households ($N$), the interest rate for liquid assets ($r$), and the cumulative number of livestock held by all households, represented as $\sum_{i=1}^{N} L_{it}$. The $p_t$ in Eq (7) is simultaneously determined in Eqs (2) and (4) based on the livestock at $t$. With the livestock price established, each individual household calculates its livestock demand (or supply) based on its herd size, as expressed in Eq (6). It is important to note that both $X_{it}$ (livestock demand or supply) and $p_t$ (livestock price) are contingent on the number of livestock owned by each household ($L_{it}$). Also, $\alpha$ included in both equations is the proportion of output produced from livestock input in the single input production model in Eq (1). The relative herd size among households becomes a crucial factor. For instance, if all households possess an equal number of livestock, there will be no livestock trading. This arises from the condition where $L_{it} = L$ for all households, resulting in $p = \alpha N/rNL = \alpha/rL$ and subsequently $X = \alpha/r(\alpha/rL) - L = 0$.

## Livestock dynamics with SIR model

Livestock population dynamics are influenced by births, deaths, and the occurrence of infectious diseases. Livestock can be categorized into susceptible (S), infectious (I), and recovered (R) groups:

$$L_{it} = S_{it} + I_{it} + R_{it}. \tag{8}$$

and each component is characterized by the following differential equations [29–32]:

$$\frac{dS}{dt} = \mu_B S_{it} - \mu_D S_{it} - \beta S_{it} I_{it}, \tag{9}$$

In these equations, $\mu_B$ (where $\mu_B$ is in the range of (0,1]), represents the fertility rate of livestock, $\mu_D$ (where $\mu_D$ is in the range of (0,1]), represents the mortality rate of livestock, and $\beta$ (where $\beta$ is in the range of (0,1]), represents the probability of a susceptible (S) livestock becoming infected upon contact with an infectious (I) one. The increase in the susceptible population is solely due to newborn livestock ($\mu_B S_{it}$), while the decrease occurs through natural mortality ($\mu_D S_{it}$) and through transitioning to the infectious status by coming into contact with infectious livestock ($\beta S_{it} I_{it}$).

$$\frac{dI}{dt} = \beta S_{it} I_{it} - \gamma I_{it} - \mu_D I_{it} - \omega I_{it}, \tag{10}$$

where $\gamma \in (0, 1]$ and $\omega \in (0, 1]$ are the recovery rate (cure) from the infectious disease and the death rate from the disease, respectively. As transferred from susceptible status ($\beta S_{it} I_{it}$), infectious status increases and decreases by disease ($\omega I_{it}$) or natural death ($\mu_D I_{it}$) and transformation to recovered status by natural healing ($\gamma I_{it}$).

$$\frac{dR}{dt} = \gamma I_{it} - \mu_D R_{it} \tag{11}$$

There is no consideration of treatment or disease control for an infectious disease. In other words, livestock infected with a disease will naturally recover from the infectious state with a certain probability ($\gamma I_{it}$) or experience natural mortality ($\mu_D R_{it}$). Additionally, the recovered equation (Eq (11)) assumes a permanent immunity for cured animals. For instance, if cattle become infected with an infectious disease and subsequently recover without dying, they are assumed to be immune to the same disease for the remainder of their lives. In this case, once livestock recover, they are not susceptible to reinfection with the same disease, and death from the same infectious disease after recovery is not considered.

Take into account the net livestock acquisition ($X_{it}$) from other households in the SIR process. Let $\theta_l^X$ represent the fraction of the status $l$ (where $l$ can be S, I, or R) in the net livestock acquisition:

$$\theta_S^X = \frac{X^S}{X}, \quad \theta_I^X = \frac{X^I}{X}, \quad \theta_R^X = 1 - \theta_S^X - \theta_I^X \quad \text{where } X \equiv X^S + X^I + X^R.$$

The subscripts $i$ and $t$ for $X_{it}$ and $\theta_{l,it}^{X_{it}}$ are omitted for simplicity. Consequently, the SIR differential equations, which are Eqs (9), (10), and (11), can be reformulated as functions of the net livestock acquisition:

$$\frac{dS(X)}{dt} = \mu_B(S + \theta_S^X X) - \mu_D(S + \theta_S^X X) - \beta(S + \theta_S^X X)(I + \theta_I^X X) \tag{12}$$

$$\begin{aligned}\frac{dI(X)}{dt} = {}& \beta(S + \theta_S^X X)(I + \theta_I^X X) - \gamma(I + \theta_I^X X) - \mu_D(I + \theta_I^X X) \\ & -\omega(I + \theta_I^X X)\end{aligned} \tag{13}$$

$$\frac{dR(X)}{dt} = \gamma(I + \theta_I^X X) - \mu_D(R + \theta_R^X X). \tag{14}$$

## Social impact measurement

Social impacts are evaluated based on two key criteria: aggregate impact and relative impact. Aggregate impact encompasses changes in the overall number of livestock and total consumption within society. On the other hand, relative impact focuses on disparities in livestock ownership and wealth among individual households. Household wealth, denoted as $W_{it}$, is calculated as the sum of the current market value of livestock ($p_t L_{it}$) and liquid assets ($A_{it}$).

Initially, the social impacts resulting from livestock diseases can be determined by aggregating the effects on individual households. These individual household impacts can be assessed by changes in livestock numbers and alterations in consumption. By analyzing the fluctuations in the overall livestock count and the consumption patterns of individual households, I can derive the direct impact on a livestock-dependent society.

1. Total Livestock: $\sum_{i=1}^{N} L_{iT}$.

2. Total Consumption: $\sum_{i=1}^{N} C_{iT} = \sum_{i=1}^{N} [\alpha \ln(L_{it} + X_{it}) + rA_{it}]$

The second approach involves an indirect measurement that compares impacts across households within the livestock-dependent society. This is achieved by calculating Gini coefficients for both livestock numbers and wealth (liquid assets) among individual households. These relative social impact assessments for different scenarios can reveal how the disease has influenced inequality across society.

3) Gini Coefficient: $\sum_{i=1}^{N} \sum_{i'=1}^{N} \mid y_{iT} - y_{i'T} \mid / 2 \sum_{i=1}^{N} \sum_{i'=1}^{N} y_{i'T}$ where $i \neq i' \in \{1, 2, \cdots, N\}$ and $y \in \{A, L, W\}$.

From the theoretical discussions conducted thus far, I have determined the trade volume chosen by households, established the equilibrium livestock price, and obtained measurements for assessing the impact of trade in the context of agent-based modeling (ABM) simulations. However, two aspects have not been explicitly addressed in this section. First, I will explore the theoretical conditions under which disease transmission can occur through trade. Second, I will look at the mechanisms governing the health status of the livestock being traded. These two points will be covered in the following sections.

## Theoretical applications

In this section, I will elaborate on the conditions for the spread of infectious diseases concerning herd size. Given that the net purchase ($X$) and the equilibrium price are contingent upon the livestock count ($L$), the relative size of herds among households becomes a crucial factor in determining both $X$ and $p$. Depending on the relative size of herds in society, the conditions under which households without infected livestock engage in the purchase of infectious livestock change. The profitability conditions that drive trade with infectious livestock are defined by the herd size.

This section will outline the disease spread conditions in which diseased livestock can disseminate within society through livestock trading, and it will consider three different scenarios:

1. Homogeneous Households: In this scenario, households are uniform, except for their initial herd sizes. For example, I assume that the fertility and mortality rates of livestock are consistent with the conventional SIR model ($\mu_B = \mu_D$).

2. Two Herd Sizes: Here, I explore scenarios where there are only two different herd sizes among households.

3. $M(\leq N)$ Herd Sizes: This scenario extends to the case where there are multiple (up to $M$) distinct herd sizes within the population.

The objective is to elucidate the disease spread conditions under these three scenarios, where the characteristics of households are primarily distinguished by their initial herd sizes. This analysis will provide insights into how different herd size distributions impact the spread of sick livestock within the society.

### Homogeneous herd size

In the most straightforward scenario, all $N$ households possess an identical number of livestock ($L_i = \bar{L}$). As a consequence, at the initial stage ($t = 0$), when the livestock counts in all

households are equivalent, there are no livestock transactions occurring between households.

$$p_0 = \frac{\alpha N}{r N \bar{L}} = \frac{\alpha}{r \bar{L}} \qquad \text{and} \qquad X_{i0} = \frac{\alpha}{\alpha / \bar{L}} - \bar{L} = 0$$

The absence of transactions at the initial stage is a phenomenon that not only occurs in the scenario with homogeneous herd sizes but also in other cases. This occurs because, under the assumption of $\mu_B = \mu_D$, livestock adjustments in accordance with the SIR model are depicted through livestock losses in $\lambda$ households with infected animals, totaling $\eta$, starting from $t = 1$. Here, $\lambda$ represents the proportion of households with infected animals, and $\eta$ signifies the proportion of infected animals within those households ($I = \eta L$) at the initial stage. Therefore, during the initial stage, irrespective of the presence of a disease, both types of households proceed to the next stage without engaging in any transactions.

Then let us consider the situation at $t = 1$. In the non-infectious groups, the livestock count remains constant because the entire livestock composition is in the susceptible state. However, within the infectious groups, the livestock count in the infectious group decreases in accordance with the disease-induced mortality rate. Given that the proportion of households with sick animals is $\lambda$, and the constant proportion of those being infectious is $\eta$, the adjusted total livestock count at $t = 1$ is as follows:

$$\sum_{i=1}^{N} L_{i1}(\lambda) = (1 - \lambda)S + \lambda\left(I + \frac{dI}{t} + \frac{dR}{t}\right)$$
$$= (1 - \lambda)N\bar{L} + \lambda[1 - \eta(\mu_D - \omega)]N\bar{L}$$
$$= [1 - \lambda\eta(\mu_D + \omega)]N\bar{L}$$

and consequently the price at $t = 1$ is

$$p_1 = \frac{\alpha}{r[1 - \lambda\eta(\mu_D + \omega)]\bar{L}}.$$

Using superscripts $\lambda$ and $1 - \lambda$ to denote the 'infectious group' and 'non-infectious group' respectively, the livestock exchange volumes and their signs are as follows:

$$X_{i1}^{1-\lambda} = [1 - \lambda\eta(\mu_D + \omega)]\bar{L} - \bar{L} = -\lambda\eta(\mu_D + \omega)\bar{L} < 0 \ \text{(Seller)}$$
$$X_{i1}^{\lambda} = [1 - \lambda\eta(\mu_D + \omega)]\bar{L} - [1 - \eta(\mu_D + \omega)]\bar{L} = (1 - \lambda)\eta(\mu_D + \omega)\bar{L} > 0 \ \text{(Buyer)}$$

This signifies that non-infectious households, which do not have livestock affected by the disease at the initial stage, act as suppliers in the livestock trade. Conversely, because their herd size is influenced by the disease, infectious households become consumers in this context.

In this homogeneous society, the transmission of infectious disease from households with infected livestock to households without infection through livestock trading is not possible. Households with infected livestock are consistently in the position of buying livestock ($X_i^{1-\lambda} < 0$), and households without infection are consistently in the position of selling livestock ($X_i^{\lambda} > 0$). This dynamic persists in subsequent stages ($t + 1$, where $t \geq 1$). Even if excess demand was met at stage $t$, in instage $t + 1$, excess demand arises again due to livestock deaths caused by the disease. Nevertheless, at this point, the initially non-infectious group with solely susceptible livestock continues to act as a livestock seller without incurring livestock losses.

## Heterogeneous herd size: Two type case

Now suppose a scenario where, unlike the homogeneous case, there are two distinct types of households. I denote $L^b$ as the size of the large initial herd, $L^s$ as the size of the small initial

herd, and $\delta$ (where $\delta$ is in the range of (0, 1]) represents the proportion of households with $L^b$ in the society.

The total livestock count owned by the society in the initial period, taking into account the excess demand resulting from livestock deaths, evolves as follows:

$$
\begin{aligned}
\sum_{i=1}^{N} L_{i1}(\lambda, \delta) &= \delta[1 - \lambda\eta(\mu_D + \omega)]NL^b + (1 - \delta)[1 - \lambda\eta(\mu_D + \omega)]NL^s \\
&= [\delta L^b + (1 - \delta)L^s][1 - \lambda\eta(\mu_D + \omega)]N
\end{aligned}
$$

The initial line represents the total livestock count at $t = 1$, accounting for losses due to infectious disease in both types of households, $L^b$ and $L^s$. In the first term on the right side of the initial line, $\delta NL^b$ signifies the total livestock count in households with $L^b$ in the initial stage, and $\lambda\eta(\mu_D + \omega)$ indicates the rate of livestock loss due to the SIR dynamics at $t = 1$. The second term pertains to households with $L^s$ (with a proportion of $(1 - \delta)$). In summary, the total livestock count at $t = 1$ can be simplified as the initial livestock count ($[\delta L^b + (1 - \delta)L^s]N$) minus the losses incurred, taking into consideration the constant livestock loss rate ($\lambda\eta(\mu_D + \omega)$).

The total livestock count at $t = 1$, denoted as $[\delta L^b + (1 - \delta)L^s]N$, is used to calculate the price at $t = 1$, which is given as $p_1 = \alpha/r[\delta L^b + (1 - \delta)L^s][1 - \lambda\eta(\mu_D + \omega)]$. This enables us to compute the trading volume based on the categories of infectious and non-infectious livestock groups (represented by the superscripts $\lambda$ and $1 - \lambda$, respectively) and big and small livestock groups (indicated by the superscripts $\delta$ and $1 - \delta$, respectively).

$$
X_{i1}^{1-\lambda,\delta} = -(1 - \delta)(L^b - L^s) - \lambda\eta(\mu_D + \omega)[\delta L^b + (1 - \delta)L^s] < 0 \text{ (Always Seller)}
$$

In the case of households without any infectious livestock and with a large initial herd size, they consistently act as sellers, offering livestock for sale in the first stage. When there is no loss of livestock due to disease (represented by $1 - \lambda$), and the number of livestock is relatively substantial (represented by $\delta$), these households $(1 - \lambda, \delta)$ invariably supply livestock to households that either experience livestock losses or possess a relatively smaller number of livestock, resulting in excess demand. As a result, there is no possibility of disease transmission arising from the acquisition of infectious livestock.

In contrast to the constant supplier scenario, the positions adopted by each group in livestock transactions at $t = 1$ vary for the remaining groups, depending on the disparities in size between the B and S groups and the parameters associated with the livestock loss rate. The net purchases for these groups, denoted as $X^{1-\lambda,1-\delta}$, $X^{\lambda,\delta}$, and $X^{\lambda,1-\delta}$, are outlined below, including the conditions that determine the sign of $X$:

- $X^{1-\lambda,1-\delta}$: Households with no infectious livestock and a small initial herd size.

- $X^{\lambda,\delta}$: Households with infectious livestock and a large initial herd size.

- $X^{\lambda,1-\delta}$: Households with infectious livestock and a small initial herd size.

These net purchases represent the volume and direction of livestock transactions for these groups and are determined based on their specific characteristics and the underlying

parameters.

$$X_{i1}^{1-\lambda,1-\delta} = \delta(L^b - L^s) - \lambda\eta(\mu_D + \omega)[\delta L^b + (1-\delta)L^s]$$

$$\begin{cases} \leq 0 \quad \text{if } \lambda \geq \dfrac{\delta(L^b - L^s)}{\eta(\mu_D + \omega)[\delta L^b + (1-\delta)L^s]} \\[4mm] > 0 \quad \text{if } \lambda < \dfrac{\delta(L^b - L^s)}{\eta(\mu_D + \omega)[\delta L^b + (1-\delta)L^s]} \end{cases}$$

$$X_{i1}^{\lambda,\delta} = -(1-\delta)(L^b - L^s) + \eta(\mu_D + \omega)[L^b - \lambda(\delta L^b + (1-\delta)L^s)]$$

$$\begin{cases} \leq 0 \quad \text{if } \lambda \geq \dfrac{L^b}{\delta L^b + (1-\delta)L^s} \\[4mm] > 0 \quad \text{if } \lambda < \dfrac{\eta(\mu_D + \omega)L^b - (1-\delta)(L^b - L^s)}{\eta(\mu_D + \omega)[\delta L^b + (1-\delta)L^s]} \\[4mm] \text{and} \quad \delta > 1 - \eta(\mu_D + \omega) \end{cases}$$

$$X_{i1}^{\lambda,1-\delta} = \delta(L^b - L^s) + \lambda\eta(\mu_D + \omega)[L^b - \lambda(\delta L^b + (1-\delta)L^s)]$$

$$\begin{cases} \leq 0 \quad \text{if } \lambda \geq \dfrac{\eta(\mu_D + \omega)L^s + \delta(L^b - L^s)}{\eta(\mu_D + \omega)[\delta L^b + (1-\delta)L^s]} \\[4mm] > 0 \quad \text{if } \lambda < \dfrac{L^s}{\delta L^b + (1-\delta)L^s} \end{cases}$$

In comparison to the initially homogeneous households' scenario, there was one group where net purchases were negative, but apart from this, the net purchase outcomes for the remaining groups were established based on specific conditions. In essence, when the proportion of households impacted by SIR dynamics ($\lambda$), in addition to their initial livestock counts, is substantial, the likelihood of transactions between different groups increases. One consistent aspect is that the group with the largest initial livestock counts and no livestock loss never engages in the purchase of livestock.

Transactions between the three groups, excluding the Always Seller group, are determined by the values of $\lambda$. Based on the terms involving $\lambda$, there is only one potential transaction scenario: $X_{i1}^{\lambda,\delta}$ being negative (seller) and $X_{i1}^{1-\lambda,1-\delta}$ being positive (buyer), as stated in Proposition 1. This specific case highlights the dynamics of livestock exchange between these groups.

**Proposition 1** *Taking into account the ranges of $\lambda$ within $X^{1-\lambda,1-\delta}$, $X^{\lambda,\delta}$, and $X^{\lambda,1-\delta}$, transactions among the three groups, excluding the Always Seller group, are feasible only when the conditions $X_{i1}^{\lambda,\delta} < 0$ and $X_{i1}^{1-\lambda,1-\delta} > 0$ are satisfied.*

The sole pathway for the spread of infectious disease occurs when the following condition is met:

**Proposition 2 (Infectious Disease Spread Condition—Two Type)** *An infectious disease spread through livestock transactions between the $(\lambda, \delta)$ group (where $X_{i1}^{\lambda,\delta} < 0$) and the $(1 - \lambda, 1 - \delta)$ group (where $X_{i1}^{1-\lambda,1-\delta} > 0$) occurs when the following conditions are met: $\eta(\mu_D + \omega) < \delta$ and $\frac{L^s}{L^b} < 1 - \frac{\eta}{\delta}(\mu_D + \omega)$.*

The conditions outlined in Proposition 2 confirm the possibility of disease transmission occurring through livestock trading driven by economic incentives when there are relative disparities in livestock size within a livestock-dependent society. If these conditions are met, all groups except the 'Always Seller' group will be affected by the disease, and in all trades occurring after stage 2, the remaining groups, except for that specific group, will experience

increased demand for livestock due to the disease. Consequently, a more significant reduction in the number of susceptible livestock in society can be anticipated compared to the homogeneous case. However, it's important to note that the extent of this reduction depends on the information completeness of each society member and individual characteristics.

## Heterogeneous herd size: M-types

The basic two-type case demonstrated that a heterogeneous herd size society is relatively less susceptible to the spread of infectious disease through livestock trading compared to a homogeneous herd size society. To generalize this, let's consider a scenario where there are households of M types categorized based on their initial herd size, such as:

$$L^1 < L^2 < \cdots < L^M, M \leq N$$

and the proportion of each type $m \in 1, 2, \cdots, M$ is represented as $\delta^m$, with the constraint that the sum of these proportions equals 1, i.e., $\sum_{m=1}^{M} \delta^m = 1$.

As in the earlier discussions, because no trading takes place at the initial stage, I can determine the total livestock count at time $t = 1$ and the corresponding equilibrium price as follows:

$$\sum_{i=1}^{N} L_{i1} = \sum_{m=1}^{M} \delta^m L^m N[1 - \lambda\eta(\mu_D + \omega)]$$

Similar to the two-type case, the total livestock count ($\sum_{i=1}^{N} L_{i1}$) at $t = 1$ is calculated by subtracting the livestock losses, considering the constant livestock loss rate, from the initial livestock size ($\sum_{m=1}^{M} \delta^m L^m N$).

$$p_1 = \frac{\alpha}{r[1 - \lambda\eta(\mu_D + \omega)] \sum_{m=1}^{M} \delta^m L^m}$$

The equilibrium price is determined by substituting $\sum_{i=1}^{N} L_{i1}$ into Eq (7).

By utilizing these values and plugging them into Eq (6), livestock purchases or sales for each $m$-type ($M - j$ where $j = 0, 1, 2, \cdots, M - 1$) can be computed for both non-infectious livestock ($\lambda$) and infectious livestock ($1 - \lambda$) groups.

$$
\begin{aligned}
X_{i1}^{1-\lambda, M-j} &= [1 - \lambda\eta(\mu_D + \omega)]\sum_{m=1}^{M} \delta^m L^m - L^{M-j} \\
&= -[1 - \lambda\eta(\mu_D + \omega)]\left\{ \sum_{m=1}^{M-j-1}(L^{M-j} - L^m)\delta^m + \sum_{m=1}^{M-j} \delta^m L^m - \sum_{m=1}^{M} \delta^m L^m \right\} \\
&\quad - \lambda\eta(\mu_D + \omega)L^{M-j} \\
X_{i1}^{\lambda, M-j} &= [1 - \lambda\eta(\mu_D + \omega)]\sum_{m=1}^{M} \delta^m L^m - [1 - \eta(\mu_D + \omega)]L^{M-j} \\
&= -[1 - \lambda\eta(\mu_D + \omega)]\left\{ \sum_{m=1}^{M-j-1}(L^{M-j} - L^m)\delta^m + \sum_{m=1}^{M-j} \delta^m L^m - \sum_{m=1}^{M} \delta^m L^m \right\} \\
&\quad + (1 - \lambda)\eta(\mu_D + \omega)L^{M-j}
\end{aligned}
$$

The pattern observed in the previous examples reveals that households with the largest livestock holdings ($L^M$) and no livestock losses due to infectious disease ($1 - \lambda$) consistently assume the role of suppliers. This proposition holds true in the more general scenario where multiple M-types exist. Regardless of the specific number of M-types, households with the

highest livestock counts and no disease-related losses are typically the suppliers in the livestock trade.

$$
\begin{aligned}
X_{i1}^{1-\lambda,M} &= -[1 - \lambda\eta(\mu_D + \omega)]\sum_{m=1}^{M-1}(L^M - L^m)\delta^m - \lambda\eta(\mu_D + \omega)L^M \\
&= -\sum_{m=1}^{M-1}(L^M - L^m)\delta^m + \lambda\eta(\mu_D + \omega)\left\{ L^M\left[\sum_{m=1}^{M-1}\delta^m - 1\right] - \sum_{m=1}^{M-1}L^m\delta^m \right\} \\
&< 0 \ \text{(Always Seller)} \\
&\quad \left( \because L^M > L^m \ \text{where} \ m = 1,2,\cdots,M-1 \ \text{and} \ \sum_{m=1}^{M-1}\delta^m - 1 < 0 \right)
\end{aligned}
$$

The scenario with the most significant risk of disease spread due to household production-based livestock trading, based on the observations from the previous two-type case, occurs when there is livestock exchange from the group with the highest initial livestock count ($L^M$), including livestock with the disease ($\lambda$), to the group with susceptible livestock of $L^{M-1}$ without the disease ($1 - \lambda$). In stage 1, the second-largest household without the disease is likely to acquire livestock from households with infectious livestock. If this condition is met, the other households, except for the group with the healthiest livestock, cannot exclude the possibility that livestock carrying the disease will enter their households.

Let us begin by examining the condition under which the group denoted as $(1 - \lambda, M - 1)$ will take on the role of a buyer at $t = 1$.

$$
\begin{aligned}
X_{i1}^{1-\lambda,M-1} &= -[1 - \lambda\eta(\mu_D + \omega)]\sum_{m=1}^{M-2}(L^{M-1} - L^m)\delta^m - \lambda\eta(\mu_D + \omega)L^{M-1} \\
&\quad +[1 - \lambda\eta(\mu_D + \omega)]\delta^M L^M \\
&= -\sum_{m=1}^{M-2}(L^{M-1} - L^m)\delta^m + \lambda\eta(\mu_D + \omega)\left\{ L^{M-1}\left[\sum_{m=1}^{M-2}\delta^m - 1\right] - \sum_{m=1}^{M-2}L^m\delta^m \right\} \\
&\quad +[1 - \lambda\eta(\mu_D + \omega)]\delta^M L^M
\end{aligned}
$$

$$
\begin{cases}
\leq 0 \ \text{if} \ \lambda \geq \dfrac{\sum_{m=1}^{M-2}(L^M - L^m)\delta^m - \delta^M L^M}{\eta(\mu_D + \omega)[\sum_{m=1}^{M-2}(L^M - L^m)\delta^m - \delta^M L^M - L^{M-1}]} \\[4ex]
> 0 \ \text{if} \ \lambda < \dfrac{\sum_{m=1}^{M-2}(L^M - L^m)\delta^m - \delta^M L^M}{\eta(\mu_D + \omega)[\sum_{m=1}^{M-2}(L^M - L^m)\delta^m - \delta^M L^M - L^{M-1}]}
\end{cases}
$$

**Proposition 3** *If* $\lambda < \frac{\sum_{m=1}^{M-2}(L^M - L^m)\delta^m - \delta^M L^M}{\eta(\mu_D + \omega)[\sum_{m=1}^{M-2}(L^M - L^m)\delta^m - \delta^M L^M - L^{M-1}]}$, *then* $X_{i1}^{1-\lambda,M-1} > 0$.

Following that, I can establish the condition under which the group with the largest infectious livestock ($\lambda, M$) will assume a supplier position in livestock trade at $t = 1$ in a similar

manner.

$$
\begin{aligned}
X_{i1}^{\lambda,M} &= -[1-\lambda\eta(\mu_D+\omega)]\sum_{m=1}^{M-1}(L^M-L^m)\delta^m + (1-\lambda)\eta(\mu_D+\omega)L^M \\
&= -\left[\sum_{m=1}^{M-1}(L^M-L^m)\delta^m + \lambda\eta(\mu_D+\omega)L^M\right] \\
&\quad +\eta(\mu_D+\omega)\left[\lambda\sum_{m=1}^{M-1}(L^M-L^m)\delta^m + L^M\right] \\
&\begin{cases}
\leq 0 & \text{if } \lambda \geq \dfrac{\sum_{m=1}^{M-1}(L^M-L^m)\delta^m - \eta(\mu_D+\omega)L^M}{\eta(\mu_D+\omega)[\sum_{m=1}^{M-1}(L^M-L^m)\delta^m - L^M]} \\[2em]
> 0 & \text{if } \lambda < \dfrac{\sum_{m=1}^{M-1}(L^M-L^m)\delta^m - \eta(\mu_D+\omega)L^M}{\eta(\mu_D+\omega)[\sum_{m=1}^{M-1}(L^M-L^m)\delta^m - L^M]}
\end{cases}
\end{aligned}
$$

**Proposition 4** *If* $\lambda < \frac{\sum_{m=1}^{M-1}(L^M-L^m)\delta^m-\eta(\mu_D+\omega)L^M}{\eta(\mu_D+\omega)[\sum_{m=1}^{M-1}(L^M-L^m)\delta^m-L^M]}$, *then* $X_{i1}^{\lambda,M} \leq 0$.

By taking into account the conditions related to the range of $\lambda$ that result in $X_{i1}^{1-\lambda,M-1}$ being positive and $X_{i1}^{\lambda,M}$ being negative, as stated in Propositions 3 and 4, I can determine the condition that fulfills the potential for the spread of infectious livestock throughout society at $t = 1$), with the exception of the constant suppliers:

**Corollary 1 (Infectious Disease Spread Condition—M-Types)** *If* $\sum_{m=1}^{M-1}(L^M-L^m)\delta^m >$ $L^M$ *and* $\eta(\mu_D+\omega) > \frac{L^M\sum_{m=1}^{M-2}(L^{M-1}-L^m)\delta^m-L^{M-1}\sum_{m=1}^{M-1}(L^M-L^m)\delta^m-\delta^M L^{M2}}{L^M\sum_{m=1}^{M-2}(L^{M-1}-L^m)\delta^m-L^M L^{M-1}-\delta^M L^{M2}}$ *then infectious livestock can potentially spread through livestock transactions occurring at t = 1 between the members of the* $(1 - \lambda, M - 1)$ *group and the* $(\lambda, M)$ *group.*

The applicability of Corollary 1 is influenced by a wide range of parameters and the number of livestock owned by households, which makes it challenging to determine specific possibilities. Even if the strict conditions outlined in the corollary are not met, it can be observed through scenarios involving two types of households and multiple types of households that in heterogeneous societies, households without disease-infected livestock are at a heightened risk of purchasing infected livestock, particularly in extreme cases. This probability arises from the inherent characteristics of such diverse communities. The mathematical proof of this possibility is intricate and depends on various factors, including the composition of the livestock owned by sellers and the resulting variations in the overall livestock population within the society.

To tackle these complexities and overcome the limitations, the agent-based modeling (ABM) simulation method is employed. ABM enables the exploration of the long-term consequences of livestock trading under scenarios where the conditions specified in Corollary 1 are met, and random partner matching for trading occurs over subsequent periods ($t > 1$). This modeling approach formally defines the behavior of each agent (households) within a multi-agent system and allows for the matching of trading partners without imposing rigid constraints.

However, it's essential to understand that due to the stochastic nature of trading partner selection, it is not possible to determine long-term equilibriums theoretically. Consequently, this section illustrates how the social implications of disease outbreaks are influenced by the randomness of trading partners, making it challenging to derive long-term effects theoretically

and underscoring the need for simulation methods. Another critical factor affecting agent behavior in the ABM simulation is $\theta_I^X$, which determines how livestock are chosen for trading based on their health status.

## Livestock selection for trading

In inter-household livestock transactions where buyers are unaware of the health status of sellers' herds, the risk of infectious disease transmission from sellers to buyers depends on the criteria that sellers employ when selecting animals for trading. When sellers do not have information about the individual health status of animals in their herd, they typically choose animals for sale without taking their health condition into account.

However, if sellers possess perfect knowledge of the health status of their herd, the patterns of disease transmission will be influenced by how sellers decide which animals from their herd to make available for sale. This knowledge can significantly impact the likelihood of disease spread through trading.

Table 1 presents the criteria used by sellers to select the proportion of sick animals to include among the total number of animals they want to sell ($\theta_I^X$). Initially, uninformed sellers opt for a random selection of the proportion of sick animals ($\theta_I^X$). In the ABM simulation, uninformed sellers use values randomly drawn from a uniform distribution. If the number of sick animals exceeds the seller's sales quantity ($X \leq I$), $\theta_I^X$ can range from 0% to 100% (Uniform [0, 1]). Conversely, if the number of sick animals is less than the seller's sales quantity, the maximum ratio of infectious animals that sellers can sell is $I/X$, so the $\theta_I^X$ range is limited to $[0, I/X]$.

Subsequently, selfish sellers give preference to trading their own infectious livestock. In cases where their sales quantity is less than the number of sick animals ($X \leq I$), $X$ consists exclusively of infectious animals, resulting in $\theta_I^X$ being set to 1. On the other hand, when the number of animals to sell exceeds the number of sick animals ($X > I$), $\theta_I^X$ becomes $I/X$, with the remaining animals being replaced by susceptible or recovered animals.

Altruistic sellers, in contrast to selfish sellers, exclusively sell susceptible livestock if the number of susceptible animals they own is greater than the quantity they intend to sell ($X \leq S$). When the number of susceptible animals is insufficient ($S + I \geq X > S$), the remaining animals are replaced with infectious livestock.

From the buyer's perspective, the situation described above becomes a concern when the trading partner possesses infected livestock. When all the livestock owned by the trading partner consist solely of healthy susceptible animals, buyers can freely engage in transactions

**Table 1. Seller's infectious livestock selection for trade by seller type.**

| Seller Type | $\theta_I^X$ |
|---|---|
| Uninformed | $\theta_I^X \sim \begin{cases} \text{Uniform } [0,1] & \text{if } X \leq I \leq L \\ \text{Uniform } \left[0, \dfrac{I}{X}\right] & \text{if } L \geq X > I \end{cases}$ |
| Selfish | $\theta_I^X = \begin{cases} 1 & \text{if } X \leq I \leq L \\ \dfrac{I}{X} & \text{if } L \geq X > I \end{cases}$ |
| Altruistic | $\theta_I^X = \begin{cases} 0 & \text{if } X \leq S \leq L \\ \dfrac{X-S}{X} & \text{if } S + I \geq X > S \quad (\text{assuing } R = 0) \end{cases}$ |

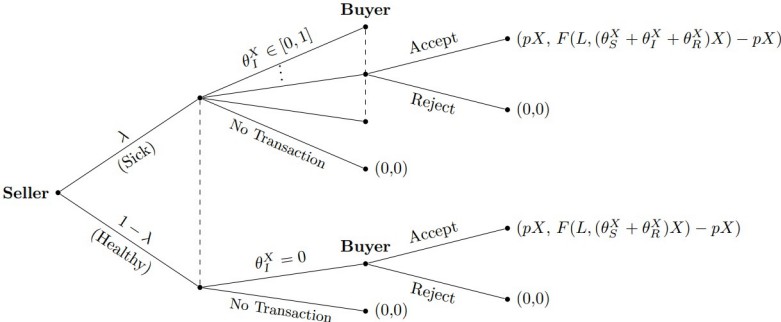

**Fig 1. Asymmetric information: Seller knows which status it is; buyer does not.**

without any worries about the trading partner's livestock selection. However, if a buyer receives only healthy animals in a livestock transaction from a specific counterpart, three possible situations may arise: 1) all the seller's livestock are healthy, 2) the seller is altruistic, or 3) the seller is uninformed and selects livestock for sale randomly, regardless of their health status. Therefore, buyers face both an imperfect information problem regarding the health status of the seller's animals and an asymmetric information problem concerning the seller's behavior. Denoting the proportion of sick animals in a household is $\eta \in [0, 1]$, Fig 1 illustrates this scenario, indicating the payoffs for both the seller and the buyer in parentheses (Seller's payoff, Buyer's payoff).

From the buyer's perspective, the situation described above raises concerns when their trading partner possesses infected livestock. When all the livestock owned by the trading partner consist solely of healthy susceptible animals, buyers can engage in transactions without any concerns about the health status of the trading partner's animals. However, when a buyer receives only healthy animals in a livestock transaction from a specific counterpart, three possible scenarios may arise:

1. All the seller's livestock are healthy.

2. The seller is altruistic and chooses to sell only healthy animals.

3. The seller is uninformed and selects livestock for sale randomly, regardless of their health status.

Consequently, buyers encounter both an imperfect information problem regarding the health status of the seller's animals and an asymmetric information problem concerning the seller's behavior. Denoting the proportion of sick animals in a household as $\eta \in [0, 1]$, Fig 1 illustrates this scenario, indicating the payoffs for both the seller and the buyer in parentheses (Seller's payoff, Buyer's payoff).

Several conclusions can be drawn from the game tree depicted in Fig 1. First, the dominant strategy for the seller is to propose a transaction to buyers regardless of the seller's herd's health information. Second, the buyer's strategies depend on their expected payoff, which is influenced by the seller's behavior and the health status of the animals offered for sale. Third, in cases where all three types of sellers (uninformed, selfish, and altruistic) coexist in a small communal society, determining the buyer's profit becomes challenging due to the information set. The vertical dotted lines in the tree represent information sets, indicating that a buyer cannot discern their position in the decision nodes connected by the dotted line. This complicates the calculation of the buyer's profit.

In this paper, for the sake of simplicity and clarity, the characteristics of all members of the society have been categorized into three types, as detailed in Table 1. These three types correspond to the following types of sellers. In all scenarios considered, buyers do not possess information about the health status of livestock, whereas sellers can belong to one of these three specified types.

1. *Uninformed Sellers*: Neither sellers nor buyers possess information about the health status of their herd or the herds of others. Uninformed sellers randomly choose animals for sale.

2. *Selfish Sellers*: Households are aware of the health status of each animal in their herd (but they are unaware of the status of others' herds). Selfish sellers prioritize the selection of infected animals for sale.

3. *Altruistic Sellers*: Households have knowledge of the health status of each animal in their herd (but they lack information about the status of others' herds). Altruistic sellers prioritize the selection of susceptible animals for sale.

The initial proportion of households with infectious livestock, denoted as λ, and the initial proportion of sick animals among those households, η (where η ranges from 0 to 1), are assumed to be known to both sellers and buyers, regardless of the seller type. This knowledge enables the calculation of the expected payoff in Fig 1.

Under the infectious disease spread conditions outlined in Proposition 2 and Corollary 1, there are incentives for all households in the society to engage in livestock transactions. As depicted in Fig 1, buyers lack information about the seller's type or the health status of individual animals offered by sellers. Therefore, if the two conditions specified in the propositions are met, buyers are willing to accept the seller's offer to trade. It is assumed that buyers cannot learn the seller's type through repeated transactions. Consequently, under the conditions for $t = 1$, buyers continue to trade irrespective of the seller's type.

Furthermore, livestock trade volumes and prices, as described in Eqs (6) and (7), are not affected by how sellers choose $\theta_I^X$. As demonstrated in Eqs (1), (2), and (8), illness influences livestock mortality but not productivity ($F(L)$). Although households who purchase sick animals may experience mortality, the households' expected profits are independent of the seller's $\theta_I^X$ in future livestock trades. In other words, the mortality experience does not influence future transactions.

The ABM simulation comprises nine distinct scenarios, categorizing society members into three groups based on relative herd size classification (homogeneous, two-type, and M-types households). Within each of these three societies, households are further subdivided based on the method of selecting $\theta_I^X$, resulting in three seller types: uninformed, selfish, and altruistic sellers. The subsequent analysis assesses the societal consequences of infectious disease transmission through livestock trading among households, taking into account these nine ABM simulation scenarios. The outcomes are evaluated based on the distribution of livestock and the trading tendencies observed among society members.

## Agent-based model simulation

In this section, I will present a comprehensive overview of the parameterization process and the steps involved in the Agent-Based Modeling (ABM) simulation. To generate results for the nine distinct scenarios mentioned earlier, I need to specify the parameters that govern household behavior, as defined by mathematical models, and determine the relative herd sizes within each society. In this context, the agents represent households, and the livestock-dependent society is characterized by the initial herd sizes held by these households. The interactions

**Table 2. Fixed parameters and initial conditions for ABM simulation.**

|  | Symbol | Description | Value |
|---|---|---|---|
| Fixed parameters | $\alpha$ | Production coefficient | 0.8 |
|  | $\beta$ | Probability of getting infectious disease in a contact | 0.02 |
|  | $\gamma$ | Recovery rate from infectious disease | 0.2 |
|  | $\omega$ | Death rate from infectious disease | 0.3 |
|  | $\mu_B$ | Baseline Fertility rate of livestock | 0.03 |
|  | $\mu_D$ | Baseline mortality rate (given no infectious disease) | 0.03 |
|  | $r$ | Interest rate for liquid assets | 0.01 |
|  | $M$ | The number of types in the $M$ herd sizes | 5 |
|  | $N$ | Total number of households ($i = 1, 2, \cdots, N$) | 100 |
|  | $T$ | Total simulation period ($t = 0, 1, 2, \cdots, T$) | 350 |
|  | $ITN$ | Total Monte-Carlo iteration ($n = 1, 2, \cdots, ITN$) | 100 |
| Initial conditions | $\lambda$ | Probability of having infectious livestock | 0.35 |
|  | $\eta$ | Proportion of the infectious livestock in herd if $I_{i,0} > 0$ | 0.3 |
|  | $\delta$ | Proportion of households with $L^b$ in two herd sizes case | 0.5 |
|  | $\delta^m$ | Proportion of households with $L^m$ in $M$ herd sizes case | 0.2 |
|  | $A_0$ | Initial liquid assets (exogenous endowment) | 500 |
|  | $R_0$ | Initial number of recovered animals | 0 |

among these agents are facilitated through actions determined by the trading mechanism, which is calculated using the general equilibrium model coupled with the SIR dynamics outlined in Eq (7).

## Parameters and herd sizes

Table 2 presents the parameterizations for fixed parameters and initial conditions in the ABM simulation. The fixed parameters are related to the household production model (specifically, $\alpha$ and $r$ in Eq (5)) and the SIR model (including $\beta$, $\gamma$, $\omega$, $\mu_B$, and $\mu_D$) in Eqs (9), (10), and (11)), and they remain consistent regardless of the livestock trading simulation. Additionally, the table includes information about the number of agents ($N$), the maximum time period ($T$), the maximum iteration ($ITN$), and the definition of $M$ in the initial herd sizes used in the ABM simulation.

The initial conditions encompass the composition of initial infectious animals ($\lambda$, $\eta$, and $R_0$), the initial proportions categorized by type of the initial herd sizes ($\delta$ for two herd sizes and $\delta^m$ for $M$ herd sizes), and the exogenous endowment ($R_0$).

The sizes of the initial herds, which reflect the characteristics of the society, are shown in Table 3 based on the household types of relative herd sizes. In Table 3, $\delta$ represents the proportion of households with $L^b$ herds. The simulation assumes $\delta = 0.5$, indicating that 50 households have 15 livestock each, while the remaining 50 households have 5 livestock each. This

**Table 3. Initial livestock herd size.**

| Household Type | Initial Livestock |
|---|---|
| Homogeneous | $\bar{L} = 10$ |
| Two herd sizes | $L^b = 15$, $L^s = 5$ ($\delta = 0.5$) |
| $M$ herd sizes | $L^1 = 5$, $L^2 = 7.5$, $L^3 = 10$, $L^4 = 12.5$, $L^5 = 15$ ($\delta^{m \in \{1,2,..,5\}} = 0.2$) |

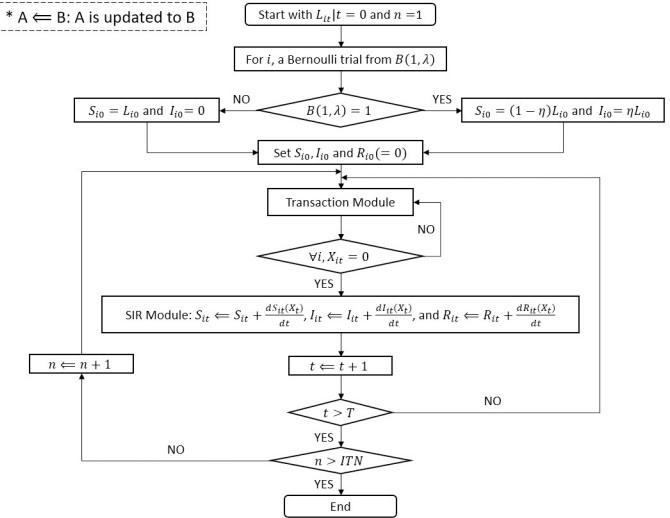

**Fig 2. ABM simulation flowchart (Transaction module is explained in Fig 3).**

configuration results in a total livestock population of 1000, which is the same as in the homogeneous case.

Additionally, $\delta^m$ denotes the proportion of herd sizes in M-type households ($m \in 1, 2, \cdots$, $M$). For the sake of comparisons between societies, this simulation assumes $M = 5$ and sets $\delta^{m \in 1,2,..,5} = 0.2$ to maintain a total livestock population of 1000. This standardization of the total livestock population across the three societies, according to the initial herd size, allows for a meaningful comparison of the long-term social impact.

## Steps for simulation

Fig 2 provides a flowchart illustrating the steps involved in the ABM simulation. The ABM simulation process includes the following key components:

- Initial Setting: This phase involves setting up the initial conditions for the simulation. It allocates the initial herd sizes defined in Table 3 to individual agents ($i = 1, 2, \cdots, N$). For instance, in the two-herd size scenario, 50 out of 100 agents have 15 livestock, while the other 50 agents have 5 livestock (where $\delta = 0.5$). At this stage, there is no distinction between the health conditions of the livestock. Each agent's initial livestock health status is determined by conducting a Bernoulli trial from $B(1, \lambda)$, which is essentially a random decision with a probability of $\lambda$ for each agent. If an agent's trial results in 1, then $\eta L$ of their livestock are classified as infectious (I status), and the remaining $(1 - \eta)L$ are classified as susceptible (S status). If an agent's trial results in 0, it means that they initially have zero infected livestock, and all of their livestock are in the susceptible state ($I = 0, S = L$). In both cases, the initial recovered livestock count is assumed to be zero ($R = 0$) at the outset (i.e., $t = 0$).

- Transaction Module: The transaction module is responsible for simulating livestock transactions between agents. This module accounts for the various seller types (uninformed, selfish, altruistic) and buyer behaviors as discussed in the previous sections.

- SIR Module: The SIR module models the dynamics of infectious disease transmission based on the SIR model equations, taking into account interactions between susceptible, infectious, and recovered livestock.

The simulation proceeds by iterating through these modules over multiple time periods ($t > 0$) to analyze the long-term social impact of livestock trading in various scenarios. The detailed flowchart in Fig 2 helps illustrate the steps of the ABM simulation process.

Fig 3 provides a detailed overview of the "Transaction Module" as part of the agent-based modeling (ABM) simulation. This module focuses on Agent $i$ as an example, but it's important to note that all agents $(1, 2, \cdots, N)$ are simultaneously engaged in this process. Here's a breakdown of the key steps and components within the "Transaction Module":

- Equilibrium Price and Net Purchase Calculation: Each agent $i$ calculates their net purchase of livestock ($X$) based on the known equilibrium price at time $t$, as described in Eq (6). This calculation takes into account the agent's initial herd size ($L$).

- Division into Seller and Buyer: Depending on the sign of their net purchase ($X$), Agent $i$ is classified as either a seller or a buyer.

- Random Matching with Opponent Agent $j$: The main agent $i$ interacts with other agents ($j \neq i$) in the transaction module. However, it's important to highlight that the assignment of agent $j$ in this context is subject to randomness to emphasize the unpredictable nature of the transactions. In this simulation, a random opponent is matched, and it's important to note that the counterpart $j$ is not consistently the same agent for each interaction.

- Meeting Trading Partners: The main objective is to match two agents, one with excess demand and another with excess supply. This process continues until the livestock trading quantity (i.e., excess demand or excess supply) of both agents reaches zero. If the trading partners don't match based on their $X$ signs, the search continues.

- Multiple Agent Matching: Agent matching occurs repeatedly until the net purchase ($X$) of all agents becomes zero, signaling the conclusion of the transaction module.

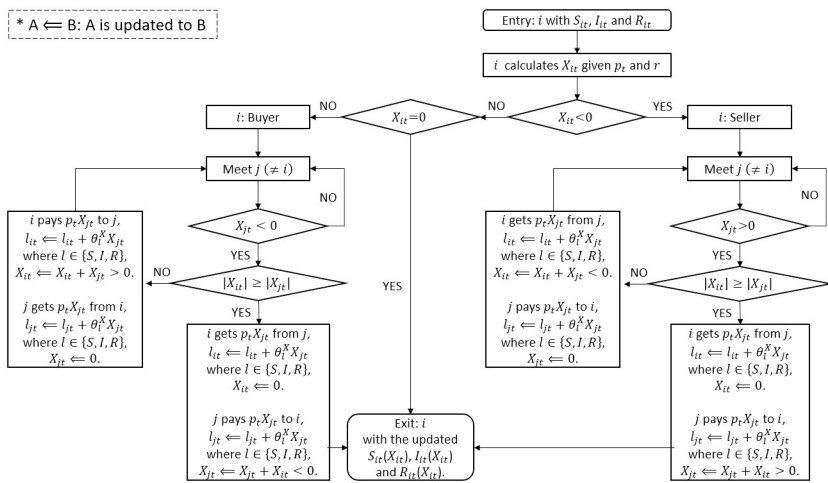

**Fig 3. Transaction module details in Fig 2.**

After the transaction module, the simulation proceeds to the SIR module. In the SIR module, livestock acquired or reduced in the transaction module is adjusted within the susceptible (S), infectious (I), and recovered (R) categories based on Eqs (12), (13), and (14). This adjustment takes into account livestock deaths, recoveries, and other factors resulting from the purchase or sale of livestock.

The simulation iterates through the transaction module and the SIR module until it reaches the specified time limit (T). This cycle repeats based on the iteration number (ITN) specified for the Monte Carlo method. Ultimately, after completing the simulation, there will be $T \times ITN$ sets of simulation results. The final simulation results are determined by calculating the average values of these results at time T. For instance, the average value of total livestock at stage T serves as a representative indicator of long-term livestock status for a given society. The actual simulation results and their implications are discussed in the subsequent section.

## ABM simulation results

Table 4 presents the results of the agent-based modeling (ABM) simulation, highlighting various social impact measures at time $t = 350$. These measures include the total number of livestock (L), consumption (C), Gini coefficients for the total number of livestock (Gini (L)), liquid assets (A), and wealth (Gini (W)).

In this table, the columns labeled Δ% in the L and C columns represent the percentage change of the long-term values at $T = 350$ compared to the initial values at $t = 0$. It's calculated as: $\Delta\%$ of $L = \frac{L_{t=350} - L_{t=0}}{L_{t=0}} \times 100\%$ and $\Delta\%$ of $C = \frac{C_{t=350} - C_{t=0}}{C_{t=0}} \times 100\%$ The "No Trade" column in the Seller Type category serves as a baseline scenario, illustrating ABM simulation results without any inter-household livestock transactions. Additionally, the term "homogeneous herd size" corresponds to the 1-type.

Fig 4 illustrates the trends over time for the total number of livestock (L) in the ABM simulation results. These trends confirm the convergence of simulation outcomes, demonstrating the stability of the results over the long term (at $t = 350$). The sub-figures in Fig 4 are organized according to the initial herd size, while the lines within each figure represent the different seller types.

The converging trends over time indicate that the simulation results stabilize as the model runs for an extended duration. The specific dynamics of convergence may vary based on the

**Table 4. ABM results at $T = 350$ ($L$, Livestock; $C$, Consumption; $A$, Liquid Assets; $W$, Wealth).**

| Initial Herd Size | Seller Type | $L(\Delta\%)$ | $C(\Delta\%)$ | Gini (L) | Gini (A) | Gini (W) |
|---|---|---|---|---|---|---|
| Homogeneous (1-type) | No Trade | 664.9 (-33.5%) | 535.2 (-21.8%) | 0.33 | 0 | 0.05 |
| | Random | 355.7 (-64.4%) | 576.5 (-15.7%) | 0 | 0.12 | 0.10 |
| | Selfish | 313.5 (-68.7%) | 580.7 (-15.1%) | 0 | 0.17 | 0.15 |
| | Altruistic | 357.0 (-64.3%) | 576.5 (-15.7%) | 0 | 0.04 | 0.03 |
| 2-type | No Trade | 672.7 (-32.7%) | 568.9 (-15.4%) | 0.47 | 0 | 0.07 |
| | Random | 345.1 (-65.5%) | 597.0 (-11.3%) | 0 | 0.15 | 0.13 |
| | Selfish | 310.4 (-69.0%) | 616.2 (-8.4%) | 0 | 0.22 | 0.19 |
| | Altruistic | 360.8 (-63.9%) | 596.1 (-11.4%) | 0 | 0.05 | 0.05 |
| 5-type | No Trade | 662.1 (-33.8%) | 547.5 (-19.3%) | 0.46 | 0 | 0.06 |
| | Random | 348.3 (-65.2%) | 586.8 (-13.5%) | 0 | 0.14 | 0.12 |
| | Selfish | 308.7 (-69.1%) | 599.8 (-11.6%) | 0 | 0.20 | 0.17 |
| | Altruistic | 358.4 (-64.2%) | 584.5 (-13.9%) | 0 | 0.05 | 0.04 |

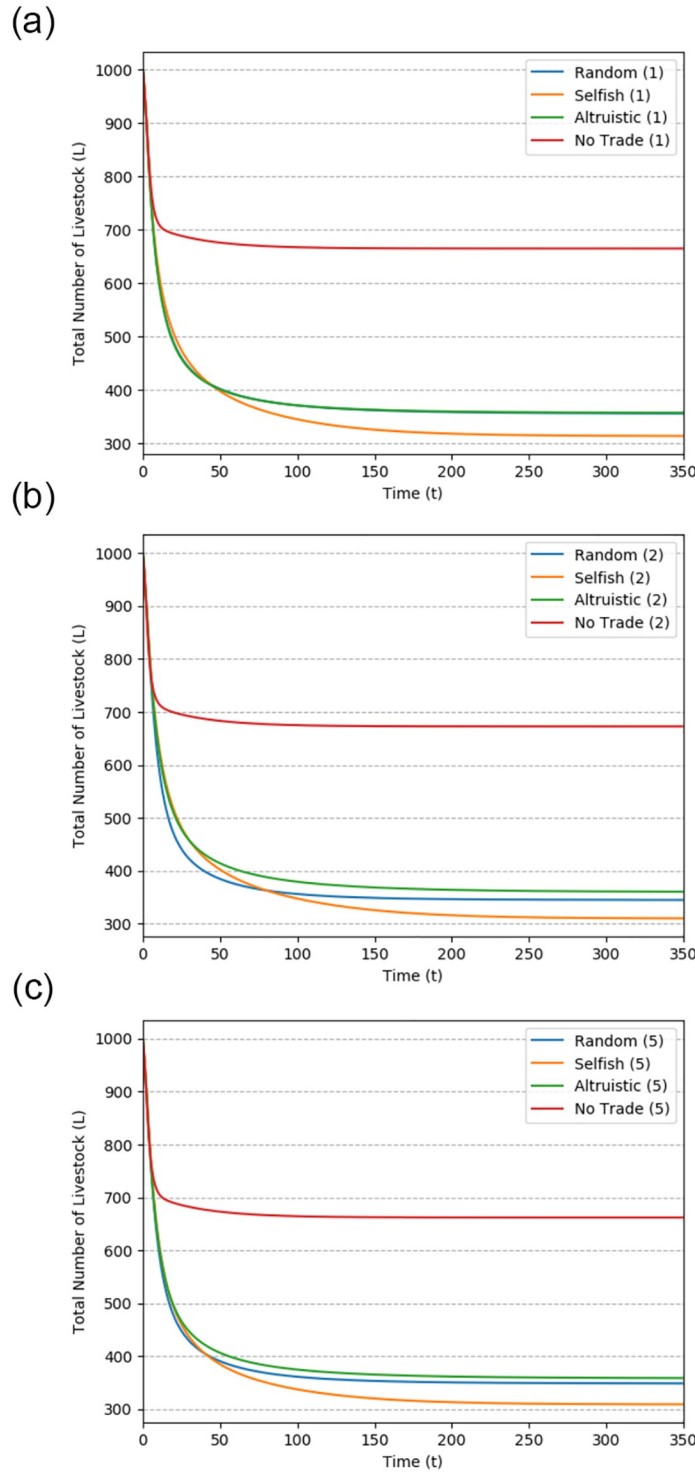

**Fig 4. Total livestock trend by initial herd size.**

initial herd size and seller type, as observed in the sub-figures. This convergence is essential to ensure that the simulation results are reliable and representative of the long-term consequences of livestock trading in various scenarios.

Across different initial herd sizes, the seller type consistently exhibits the following pattern regarding the total number of livestock in the long term:

$$\text{Total Livestock } (L) : \text{Selfish} < \text{Random} < \text{Altruistic} < \text{No Trade}.$$

Livestock trading can have a detrimental impact on the overall number of livestock within a society when compared to a scenario where livestock trading is not taking place. This is primarily due to the potential spread of diseases that can occur in conditions where profit is the main motivation for trading, as explained in Proposition 2 and Corollary 1. When livestock trading is allowed, societies consisting of selfish sellers, who intentionally preferentially use diseased livestock for trading, experience the most significant decline in the total number of livestock, with a reduction of 69.

Conversely, in societies composed of altruistic sellers who aim to restrict trading involving sick animals, the reduction in the total number of livestock is comparatively lower. Interestingly, uninformed sellers, who randomly select animals for trading, achieve results similar to those of the altruistic sellers. This outcome is related to the assumptions made in the model, particularly regarding the health status of the majority of the livestock in the society, where $\lambda \times \eta = 0.105$. In such cases, the negative effects on livestock numbers are mitigated due to the prevalence of healthy livestock in the trading pool.

On the other hand, consumption, which serves as an indicator of overall societal well-being, follows an inverse pattern to the total livestock order:

$$\text{Total Consumption } (C) : \text{No Trade} < \text{Altruistic} \leq \text{Random} < \text{Selfish}.$$

In terms of individual utility functions being determined by consumption (as shown in Eq (4)), the total utility of a society composed of selfish sellers is the highest, while the utility of societies with random sellers or altruistic sellers is relatively lower. Interestingly, in cases where livestock trading is not permitted (the "no trade" case), consumption experiences a significant decrease. It's worth noting that the level of consumption in the "no trade" case is influenced by the initial herd size, particularly when considering the percentage changes:

$$C^1 < C^5 < C^2 \text{ at } t = 350$$

where the use of superscripts indicates the type of society based on the initial herd size. This relationship highlights that the presence of variations in initial livestock sizes leads to differences in the long-term increase in consumption resulting from livestock trading when compared to a society with a homogeneous initial herd size.

The benefits of livestock trading can be determined by considering the relative reduction in livestock and the increased consumption resulting from livestock transactions. The findings from the total livestock and total consumption data reveal that livestock trading leads to increased long-term livestock losses, but it also results in higher consumption levels. Table 5 presents the gains from livestock transactions, comparing the figures for livestock loss and

**Table 5. Gains from livestock transactions (at $t = 350$).**

| Initial Herd Size | Seller Type | $\Delta L$ | $\Delta L$ Value | Total $\Delta C$ | Net Gain |
|---|---|---|---|---|---|
| Homogeneous (1-type) | Random | -309.2 | -3720 | 13110 | 9390 |
| | Selfish | -351.4 | -4227 | 15191 | 10964 |
| | Altruistic | -307.9 | -3704 | 13041 | 9337 |
| 2-type | Random | -327.6 | -3895 | 12677 | 8782 |
| | Selfish | -362.3 | -4308 | 16497 | 12189 |
| | Altruistic | -311.9 | -3709 | 11485 | 7776 |
| 5-type | Random | -313.8 | -3791 | 11108 | 7317 |
| | Selfish | -353.4 | -4269 | 14722 | 10453 |
| | Altruistic | -303.7 | -3669 | 9506 | 5837 |

increased consumption. In Table 5,

$$\Delta L_t = L_t^{\text{st}} - L_t^{\text{No Trade}}, \Delta C_t = C_t^{\text{st}} - C_t^{\text{No Trade}}$$

where $\text{st} \in \{\text{Random, Selfish, Altruistic}\}$ and Total $\Delta C = \sum_{t=0}^{t=T} \Delta C_t$.

In Table 5, $\Delta L$ and $\Delta C$ represent the changes in livestock and consumption compared to the 'No Trade' case at time $t$, respectively. 'Total $\Delta C$' signifies the overall increase in consumption in society until time $T$. The '$\Delta L$ Value' is evaluated using equilibrium prices calculated as the total livestock number of the 'No Trade' case at $t = T$, with equilibrium prices approximately equal to 12.03, 11.89, and 12.08 for the respective scenarios. The 'Net Gain' is the sum of '$\Delta L$ Value' and 'Total $\Delta C$', and as the price of the general good ($C$) is assumed to be 1, the net gain is expressed in monetary terms.

Analyzing the results for the net gain in Table 5, it is evident that there are positive gains from trade in all cases. This demonstrates that members of society experience a net benefit by increasing social well-being through enhanced consumption, even when dealing with infectious livestock.

One interesting observation among the results in Table 5 is that the social well-being, represented by 'Total $\Delta C$', and the gains from livestock trade, denoted as 'Net Gain', are the highest for selfish sellers:

$$\text{Total } \Delta C, \text{ Net Gain : Altruistic} < \text{Random} < \text{Selfish}.$$

In other words, when individuals selfishly trade sick livestock, it leads to an overall increase in livestock losses in society. However, the individual well-being, as measured by consumption, also increases. On the contrary, altruistic behavior in livestock transactions has a negative impact on social well-being.

Additionally, the relationship between this pattern of "Total $DeltaC$" and "Net Gain" does not hold true for initial herd size. The social well-being rises whether selection is random or altruistic since the initial herd size (5-type < 2-type < 1-type) does not significantly vary between households. Nonetheless, 5-type < 1-type < 2-type in a society where selfish selection is practiced. The social well-being typically reduces with time as the initial herd size discrepancies across families increase. The long-term consumption connection at $t = 350$ indicated in Table 4 ($C^1 < C^5 < C^2$) and this result are slightly dissimilar. The difference in the initial herd size between households does not significantly affect suppressing the decrease in consumption

at the beginning of the simulation, but it does cause a difference in consumption reduction in the long-term, according to a comparison of the accumulated consumption (Total $\Delta C$ in Table 5) with the consumption in the long-term ($C$ in Table 4).

If members gain from increased consumption, should they act like selfish sellers with the greatest increase in consumption, going back to the consumption results in Tables "ref"results1" and "ref"results2?" The Gini coefficient on the distribution status of wealth (Gini ($A$)) and liquid assets (Gini ($A$)) can be used to confirm the answer to this question. In this study, wealth ($W$) is the total of liquid assets ($A$) and the monetary values of livestock valued at livestock trading prices ($ptimesL$). Furthermore, the total amount of liquid assets in society remains constant because the principal is used for livestock trading and the interest income is consumed. The total amount of wealth is also unaltered: $\sum_i^N p_t \times L_{it} = \sum_i^N \frac{\alpha N}{r \sum_i^N L_{it}} \times L_{it} = \frac{\alpha N}{r}$.

The distribution status in society is then worsened by the size of the Gini ($A$) and Gini ($W$). The following summarizes the findings of the Gini coefficients:

$\sum_i^N p_t \times L_{it} = \sum_i^N \frac{\alpha N}{r \, sum_i^N L_{it}} \times L_{it} = \frac{\alpha N}{r}$. Then, the larger the Gini ($A$) and Gini ($W$), the worse the distribution status in society. The results of the Gini coefficients are summarized as follows:

$$\text{Gini Coefficient } (A) : \text{No Trade} < \text{Altruistic} < \text{Random} < \text{Selfish},$$
$$\text{Gini Coefficient } (W) : \text{Altruistic} < \text{No Trade} < \text{Random} < \text{Selfish}.$$

Since there is no asset transfer, the no trade case in Gini ($A$) is equal to zero. In a society that values altruism over profit, wealth is distributed more fairly when livestock trade is taken into account. This is because the livestock holdings of households have significantly declined as a result of the absence of livestock trading:

$$\text{Gini Coefficient } (L) \quad : \quad 0 < 1 - \text{type} < 5\text{-type} < 2\text{-type}.$$

The Gini coefficient for livestock is substantially higher than the size of liquid assets, despite variations depending on the initial herd size. The distribution of total wealth is affected by this variation in the distribution of cattle (no trade instance). The selfish society has the worst distribution status, which means that its increased consumption is slanted in favor of one particular group. There is a negligible difference when comparing the reduction in consumption of selfish vendors with that of random merchants or altruistic sellers. For instance, in the 2-type situation, the altruistic case's consumption decreases by 30.3% (1–11485/16497) in comparison to the selfish case, while the wealthiest case's Gini coefficient differs by 4 times. Therefore, selfish behavior is detrimental when taking into account the overall well-being of a society.

In summary, the ABM results imply: 1) for-profit livestock trade increases livestock losses through the trade of infectious animals between households, regardless of the type of seller. Even if the seller tries not to use infectious animals for humane purposes, the loss of livestock is still greater than if there was no commercial transaction. 2) Profits from livestock trade, valued by increased consumption, are greater than livestock losses. 3) Social welfare with increasing consumption is greatest in a selfish society in terms of quantity, but worst in terms of wealth distribution. 4) The distribution of household wealth is most equal in a society of altruistic sellers. In fact, the distribution of altruistic livestock transactions is more equitable than no trade at all. 5) The more similar the initial herd size is across households, the higher the social welfare that accrues from increased consumption.

## Discussion

In this section, I consider the implications and limitations stemming from the theoretical model and the results obtained through the Agent-Based Model (ABM) presented in this paper. One significant implication of this study is that it suggests an avenue for enhancing social well-being, even in scenarios involving the trade of highly infectious animals. In the ABM simulations, a mortality rate of 30% ($\omega$) was assumed for livestock diseases, and due to factors such as asymmetric information and the presence of profitable conditions, buyers consistently engaged in trade, even when the health status of the livestock was unknown. However, the losses incurred due to the trading of highly contagious animals were counterbalanced by the increased consumption resulting from livestock trading. This implies that an unconditional prohibition of livestock trading, which includes the potential for disease transmission, may not necessarily yield a positive impact on society as a whole.

It is important to note that the aforementioned implication is based on a highly simplified model, and generalization should be approached cautiously. The model in this study does not specify a particular disease, which means that the negative impacts on human society may vary depending on the characteristics of the disease or the specific challenges posed by infectious livestock. Furthermore, the persistent issue of asymmetric information between buyers and sellers remains unresolved, and the results could differ if buyers possessed knowledge of the health status of the traded animals. Additionally, external influences are challenging to eliminate, as the simulated society in this study represents a rural, livestock-dependent community without access to external markets. Therefore, further research is necessary to establish more comprehensive policy implications that can be generalized to broader contexts.

Another significant implication arising from the ABM results is the importance of implementing appropriate policies or actions to mitigate wealth inequality within a selfish society grappling with livestock diseases. In such a scenario, selfish sellers intentionally promote the spread of infectious animals within the society while safeguarding their own livestock from the risk of transmission, thus having a detrimental impact on social inequality. This phenomenon becomes particularly pronounced when there is a large initial herd size, and infectious animals are present. In the practical application of the theoretical model, with the exception of households categorized as 'Always sellers,' if the households with the largest number of livestock engage in trading infectious livestock, it exacerbates societal inequality. This is because, by swiftly offloading sick animals from their herds through trading, selfish sellers increase the likelihood of transmitting the disease to buyers' livestock while safeguarding their own assets. Consequently, it becomes imperative to explore policy alternatives aimed at addressing the inequality stemming from variations in initial livestock sizes and the presence of infectious animals.

This paper is not without its limitations. Firstly, the theoretical model and applications do not incorporate government interventions related to disease control. The deliberate omission of government biosecurity measures was intended to focus specifically on the impact of profit-driven incentives among households engaging in transactions involving infectious livestock. However, it's worth noting that if government-provided incentives for monitoring livestock diseases function effectively, some households might choose to contribute to the mitigation of disease spread rather than deviate from responsible practices, particularly in a heterogeneous society [21, 24, 33, 34]. In cases where government intervention is more specific and includes considerations such as indemnity design and associated costs for households, the analysis could better manage the effects of deviating households on society as a whole.

Secondly, the theoretical model and ABM simulation do not incorporate dynamic decision-making processes. The myopic decision-making approach does not account for the

potential decline in the present value or future losses of livestock due to the spread of diseases in each transaction. In cases where the risks associated with disease transmission persist, it may be essential to adequately consider these risks in the consumption component of household production. Furthermore, information such as the history of selling infectious livestock by sellers does not influence subsequent transactions in the simulation. In reality, livestock movements between households often depend on networks formed through past transactions [2, 7, 9]. Additionally, livestock trading networks may emerge based on preferences or priorities related to trading partners, such as factors like relative distance between households within a society [8], or the chronological order of interactions [4]. Therefore, it becomes imperative to incorporate dynamic decision-making processes that account for trading risks in livestock trading decisions and the dynamics of trading partner selection within the livestock trading network.

Thirdly, this paper relies on parameterizations that do not utilize specific disease-related data or data from particular regions. While the simulation approach employed in this paper offers the advantage of facilitating general discussions, it comes with the drawback of being unable to accurately estimate the actual extent of damage because intentionally chosen parameters are used. Incorporating data from specific regions could allow for the identification of diseases likely to occur in those areas and enable the simulation of resulting social damage based on trading behavior. The utilization of data parameterization has the potential to contribute to the development of policy implications in the future. And because ABM is greatly influenced by parameter choice, more realistic and rigorous consideration of the selection variables handled in the model is needed. In addition, another problem related to parameterization is that the actions and states of agents in the simulation model are too simple. For example, agents can have states such as livestock holding and liquid assets, or actions such as do not trade in addition to buy and sell.

Fourthly, the present theoretical model does not incorporate the reinvestment of wealth accumulated through livestock trading. In the ABM results, the long-term equilibrium shows a decline in the overall livestock count primarily due to the emphasis on livestock loss attributable to infectious diseases, compared to the effects of natural mortality and fertility. If households were to invest their wealth incrementally to enhance livestock breeding (e.g., through insemination services), they could anticipate a long-term reduction in livestock losses and an increase in overall wealth. Consequently, a more realistic model may involve accounting for variations in capital accumulation among households that consider reinvestment.

## Conclusion

This study developed an agent-based model (ABM) that categorized society based on initial herd size and trading behavior to assess the repercussions of infectious disease transmission within livestock-dependent communities through inter-household livestock transactions. Unlike previous research that primarily focused on livestock disease prevention and initial control, this ABM simulation aimed to shed light on the societal impacts of infectious livestock transmission driven by profit-motivated trade. The simulation results revealed that while the spread of infectious livestock through transactions led to long-term losses in livestock, it concurrently increased social welfare through enhanced consumption. Moreover, the adverse societal effects on total livestock and wealth distribution were exacerbated when livestock sellers acted selfishly. In cases where selfish livestock trading occurs within a rural society already grappling with the initial outbreak of livestock diseases, pertinent policies should prioritize identifying disease characteristics and addressing wealth inequality exacerbated by such trading, rather than outright prohibition of livestock transactions.

## Supporting information

**S1 Appendix.**
(PDF)

## Acknowledgments

This paper is based on the fourth chapter of my Ph.D. dissertation. I sincerely thank Jonathan Yoder and Eric Lofgren for their valuable guidance.

## Author Contributions

**Conceptualization:** Hyeonjun Hwang.

**Data curation:** Hyeonjun Hwang.

**Formal analysis:** Hyeonjun Hwang.

**Funding acquisition:** Hyeonjun Hwang.

**Investigation:** Hyeonjun Hwang.

**Methodology:** Hyeonjun Hwang.

**Project administration:** Hyeonjun Hwang.

**Resources:** Hyeonjun Hwang.

**Software:** Hyeonjun Hwang.

**Supervision:** Hyeonjun Hwang.

**Validation:** Hyeonjun Hwang.

**Visualization:** Hyeonjun Hwang.

**Writing – original draft:** Hyeonjun Hwang.

**Writing – review & editing:** Hyeonjun Hwang.

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
