## [Decision Letter · Decision Letter 0]

3 May 2024

PONE-D-23-36111Simulation Design to Find the Welfare Impacts of Livestock Trading and Disease TransmissionPLOS ONE

Dear Dr. Hwang,

Thank you for submitting your manuscript to PLOS ONE. After careful consideration, we feel that it has merit but does not fully meet PLOS ONE’s publication criteria as it currently stands. Therefore, we invite you to submit a revised version of the manuscript that addresses the points raised during the review process.

We look forward to receiving your revised manuscript.

Kind regards,

Junhuan Zhang, PhD

Academic Editor

PLOS ONE

Journal requirements

 [Kyungpook National University Research Fund, 2022].  

[This paper is based on the fourth chapter of my Ph.D. dissertation. This research was 1002 supported by Kyungpook National University Research Fund, 2022.]

 Kyungpook National University Research Fund, 2022]

6. We note that your Data Availability Statement is currently as follows: [All relevant data are within the manuscript and its Supporting Information files.]

Reviewers' comments:

Reviewer's Responses to Questions

**Comments to the Author**

1. Is the manuscript technically sound, and do the data support the conclusions?

Reviewer #1: Yes

Reviewer #2: Partly

2. Has the statistical analysis been performed appropriately and rigorously? 

Reviewer #1: Yes

Reviewer #2: Yes

3. Have the authors made all data underlying the findings in their manuscript fully available?

Reviewer #1: Yes

Reviewer #2: Yes

4. Is the manuscript presented in an intelligible fashion and written in standard English?

Reviewer #1: Yes

Reviewer #2: No

5. Review Comments to the Author

Reviewer #1: The article is written very rigorously, but there are the following suggestions:

1. In formula (2) of the article, it is mentioned that the goal of Agent is to maximize pi_ {it}, but in formula (4), it is mentioned again that the goal of Agent is to maximize U (C_ {it}). Please check the wording of the article;

2. When drawing, all results can be drawn instead of just showing the total number of livestock;

3. The maximum number of livestock populations set in the article is 5, and more types can be further discussed.

Reviewer #2: This study employs a model combining ABM and SIR to investigate the welfare impact of livestock trading on households, considering various types of buyers and sellers and policy analysis perspectives. However, there are several areas for improvement:

1. The abstract needs to be summarized and condensed, as it currently contains redundant information.

2. The symbol representation in this paper is highly confusing, with issues of misused, disorderly, and reused key variables.

3. The actions and states of agents may be too simplistic; additional actions and states could be considered. It might be beneficial to define the agent states (e.g., livestock holding, liquid assets) and actions (e.g., buy/sell/do not trade), agent strategies (e.g., consumption-smoothing strategy), and agent types (uninformed, selfish, altruistic) before introducing the model.

4. Is there a theoretical basis or related analysis for the production function in Equation 1?

5. For the constraint in Equation 5, since (X{i,t}) represents net purchase, the positivity or negativity of (X{i,t}) should be considered. A more detailed explanation of (X_{i,t}) at its first mention or in Equation 5 is warranted.

6. In the simulation section, explanations of variable values should be added. Different parameter choices in ABM can greatly influence simulation results.

7. Using vector graphics for the figures in the paper could enhance readability.

Additionally, there are some spelling and expression errors:

1. There's a formatting error in the quotation marks in F(L{i,t}) above Equation 1, and the notation for (L{i,t}) here differs from its form in the equation.

2. The phrase "and t" inside the parentheses on line 156 is unrelated to the context and should be removed.

3. The (X) in line 178 should match the (X_{i,t}) in Equation 2.

4. The meaning of (C_{i,t}) in Equation 3 is not introduced.

5. Some names need to be standardized; for example, "farmer" on line 200 and "farmers" on line 224 should be consistently referred to as "household" unless they have different meanings in the model, in which case they should be clearly defined and indicated in the subscripts of the symbols. Additionally, "product consumption, denoted as C" on line 205 differs from Equation 3 and should be specifically described, with the symbol's meaning made clearer.

6. (C{t}) on line 213 should be (C{it}), and the same applies to line 215. The author needs to carefully check and correct the symbol system in the paper to ensure consistency in symbol meanings. Further instances of symbol misuse are not elaborated here.

7. Both Equation 2 and Equation 5 (the equilibrium price on line 257) include (p_t), but they represent different meanings.

8. The meaning of the variable (\\alpha) in Equation 6 is not explained.

6. PLOS authors have the option to publish the peer review history of their article (what does this mean?). If published, this will include your full peer review and any attached files.

Reviewer #1: No

Reviewer #2: No

---

## [Author Response · Author response to Decision Letter 0]

23 Jul 2024

I attached the response to reviewers file to respond to the comments.

---

## [Decision Letter · Decision Letter 1]

27 Aug 2024

Simulation Design to Find the Welfare Impacts of Livestock Trading and Disease Transmission

PONE-D-23-36111R1

Dear Dr. Hwang,

We’re pleased to inform you that your manuscript has been judged scientifically suitable for publication and will be formally accepted for publication once it meets all outstanding technical requirements.

Kind regards,

Junhuan Zhang, PhD

Academic Editor

PLOS ONE

Additional Editor Comments (optional):

Reviewers' comments:

Reviewer's Responses to Questions

**Comments to the Author**

1. If the authors have adequately addressed your comments raised in a previous round of review and you feel that this manuscript is now acceptable for publication, you may indicate that here to bypass the “Comments to the Author” section, enter your conflict of interest statement in the “Confidential to Editor” section, and submit your "Accept" recommendation.

Reviewer #1: All comments have been addressed

Reviewer #2: All comments have been addressed

2. Is the manuscript technically sound, and do the data support the conclusions?

Reviewer #1: Yes

Reviewer #2: Yes

3. Has the statistical analysis been performed appropriately and rigorously? 

Reviewer #1: Yes

Reviewer #2: Yes

4. Have the authors made all data underlying the findings in their manuscript fully available?

Reviewer #1: Yes

Reviewer #2: Yes

5. Is the manuscript presented in an intelligible fashion and written in standard English?

Reviewer #1: Yes

Reviewer #2: Yes

6. Review Comments to the Author

Reviewer #1: The paper have adequately addressed the comments raised in a previous round of review. This paper is a very meaningful study, the proof process of the paper is perfect, and the simulation results accord with the theoretical expectation.

Reviewer #2: (No Response)

7. PLOS authors have the option to publish the peer review history of their article (what does this mean?). If published, this will include your full peer review and any attached files.

Reviewer #1: No

Reviewer #2: No

---

## [Editor Report · Acceptance letter]

30 Aug 2024

PONE-D-23-36111R1 

PLOS ONE

Dear Dr. Hwang, 

I'm pleased to inform you that your manuscript has been deemed suitable for publication in PLOS ONE. Congratulations! Your manuscript is now being handed over to our production team.

Kind regards, 

on behalf of

Dr. Junhuan Zhang 

Academic Editor

PLOS ONE